# Teacher Professional Development, Character Education, and Well-Being: Multicomponent Intervention Based on Positive Psychology

Diego García-Álvarez [1,2,3,*] , María José Soler [2,3] , Rubia Cobo-Rendón [4] and Juan Hernández-Lalinde [5]

1   Departamento de Ciencias del Comportamiento, Universidad Metropolitana, Caracas 1073, Venezuela
2   Centro de Estudios de Psicología, Universidad de Montevideo, Montevideo 11600, Uruguay
3   Asociación Jóvenes Fuertes Uruguay, Montevideo 11500, Uruguay
4   Facultad de Psicología, Universidad del Desarrollo, Concepción 4030000, Chile
5   Independent Researcher, Cúcuta 540001, Colombia
*   Correspondence: ddgarcia@unimet.edu.ve

**Abstract:** The COVID-19 educational crisis has generated both psychosocial risks and growth opportunities for teaching staff; these are challenges to be addressed from the perspective of sustainable development in SDG 3 Health and Well-being and SDG 4 Quality Education. During the pandemic, a character education training experience was carried out for principals and teacher coordinators, with the dual purpose of developing professional competencies for the application of positive psychology in educational centers and strengthening teacher well-being: specifically, dedication and enthusiasm in conjunction with personal resources such as self-efficacy and resilience. The multicomponent intervention based on positive psychology applied to education was carried out with a sample of 32 teaching coordinators and school principals (mean age 45.9 years; 93.75% female staff and 71.8% between 16 and 21 years of experience) from different departments in Uruguay. The results suggest that the intervention was effective, detecting higher scores in the post-test in self-efficacy ($F = 18.17$, $p < 0.001$, $\eta^2 = 0.40$), resilience ($F = 13.41$, $p = 0.001$, $\eta^2 = 0.33$), dedication and enthusiasm ($F = 8.09$, $p = 0.008$, $\eta^2 = 0.23$), and teacher training ($F = 8.36$, $p = 0.007$, $\eta^2 = 0.24$). It is concluded that the training program can provide an opportunity for improving teacher health and well-being, as well funcitoning as a device for promoting teacher professional development.

**Keywords:** Teacher professional development; character education; well-being; positive psychology

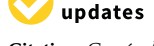



## 1. Introduction

The scientific literature has highlighted the impact of COVID-19 on the general population in various countries. It can be interpreted as a multidimensional event with implications for psychosocial trauma, expressed in increased physical and psychological symptoms such as stress, anxiety, sleep problems, increased risk behaviors such as alcohol, cigarette, and other drug use, and even a prevalence of post-traumatic stress disorder [1–5]. Therefore, COVID-19 brought multiple economic, social, health, and educational emergencies. Regarding this last aspect, schools closed abruptly. They went from a model of face-to-face educational care to a distance education model that, due to the unexpected transition generated by the health situation, could be called remote emergency education [6]. During the period when this educational modality was developed, confinement in different parts of the world was used as a way to stop the advance of the virus and the escalation of deaths. In the educational setting, it has been found that the pandemic affected student satisfaction with learning, with instructor performance being an important factor, especially with respect to student–instructor interaction, student–student interactions [7,8]. It also affected the health of students at all educational levels, from children to university students [9,10].

Research has reported the consequences of this period at the level of teacher welfare and health [11–13]. In Chile, the pandemic made clear the vulnerability of the teaching profession, emphasizing the life–work imbalance, the overwork possibly due to the rush to convert courses initially planned in a face-to-face manner to an educational modality that was being built day by day, and concerns about the mental health, well-being, and contagion of both themselves and their students. Female teachers and younger teachers were exposed to more significant risk factors at the mental health level [14]. In Portugal, teacher well-being and job satisfaction decreased during the pandemic, while concerns about future career expectations increased [15]. In Spain, it was found that teachers felt the impact of remote emergency education decreased their well-being in their emotional experience, reporting a higher frequency of negative emotions than positive ones. Difficulties in work motivation significantly impact female teachers, teachers with students of low socioeconomic status, and those in public schools [16].

In the UK, teachers described the work experience during confinement as follows: "my brain feels like a browser with 100 tabs open". Indeed, that phrase reflects the increased work demands related to uncertainty, increased workload, multiple functions, negative perception of the profession, and even decreased teacher well-being and health, which was expressed in increased anxiety, stress, and even a devaluation of the teaching profession. However, it was found that coping strategies, social support, and job autonomy are psychosocial resources that would be protective (or, at least, buffering) factors [17]. In Finland, high levels of job stress were reported in teachers which affected teachers' occupational well-being. However, job engagement was not affected; this study is an essential contribution to understanding the construct of teacher well-being in terms of levels of teacher engagement, individual experiences, and aspects related to teaching [18]. In Italy, it was reported that teachers experienced increased overall levels of burnout and distress. These psychological factors impacted the roles and perception of metacognitive processes involved in teaching, while personal accomplishment was reduced during the pandemic. It is important to note that high self-actualization was configured as a predictor of socioemotional, sociorelational, and pedagogical didactic competencies during this period. It was found that self-efficacy is a protective factor in teaching as it is a mediator between self-actualization and socioemotional competencies [19].

Similarly, in Italy, it was reported that both teachers and leaders of educational management at the center of the discipline, i.e., principals and coordinators, consider promoting health in the school necessary. However, this ideal was often overshadowed by the educational emergency, while principals experienced a high emotional price, overwhelm, the great responsibility for managing the school during the crisis, and concern for the health of the educational community. The teachers showed low to medium self-efficacy concerning the adoption of strategies for health promotion in the school. In contrast, they reported interest and motivation to do so but did not. They did not receive support from experts, nor did they receive updated materials to apply during the pandemic. One of the main recommendations arising from this educational transition during the pandemic is the development of communities and teacher support networks as a space for the exchange, reflection, and analysis of good practices both at the educational level and for teacher welfare [20].

In the Philippines, it was found that there are certain protective factors in teachers during the COVID-19 educational emergency; among them are resilience, optimism, satisfaction with both career and salary, and the subjective well-being of the teacher [21]. In Romania, general and specific self-efficacy in teaching activities, emotional management, student behavior, social support factors, career development opportunities, working conditions, or others such as school leadership have been reported to significantly affect teachers' job satisfaction and well-being [22]. In Australia, teachers reported higher levels of stress, feelings of loneliness and isolation, and low frequency of positive emotions such as joy. It is also essential to mention that teachers' well-being and teacher self-efficacy were both negatively associated with teacher stress [23]. In China, studies of teachers have reported the

importance of teacher competence for teaching, emphasizing the virtual modality that prevailed in the pandemic period, and the role of resilience in learning outcomes and teacher competence, as well as enthusiasm, self-efficacy, and well-being [24]. In our Uruguayan context, the National Evaluation Institute [25–27] reported that teachers received support from the pedagogical leadership of the principal, collective responsibility and collaboration among teaching peers, adequate internet connection, and the management of virtual tools. They reported an overload of work—especially in female teachers—in the reconciliation of work and home dynamics, and greater feelings of anguish, loneliness, fear, and overflow were reported by women than by male teachers. Difficulties were also reported at the level of accompanying the student and generating motivation for participation in virtual classes.

The studies above suggest that the design, implementation, and intervention strengthen socioemotional competencies related to teacher well-being and health. Therefore, training programs should address two levels of intervention the level of personal resources for teaching and strengthening didactic-pedagogical competencies for improving teaching and learning improving educational quality [28]. The theoretical framework assumed in this research is the theoretical explanatory model empirically generated by INEEd in its study of teachers' occupational health, carried out at the national level in Uruguay [25,26,29], which conceives well-being and teachers' health as constructs resulting from a series of complete interactions between factors of teachers' reality. In this sense, emphasis is placed on well-being as a construct that accounts for dedication and enthusiasm (elements of engagement). Furthermore, concerning teacher health, reference is made to the personal resources that act as protective factors against the different labor-related risks of the teacher's work; for example, self-efficacy, resilience, and others such as job satisfaction and meaning.

During the pandemic period, there have been some interventions aimed at teacher training, mental health, and teacher well-being which offer tools to face the challenges during and after COVID-19. One of them, carried out with Spanish teachers, focused on the reduction of teacher stress, reduction of emotional exhaustion and depersonalization, a greater sense of personal accomplishment, the development of emotional intelligence skills and the strengthening of didactic competences, the management of technological resources in the classroom, and implementation of strategies related to emotional intelligence in their teaching practice [30]. In Israel, an intervention was carried out with teachers during the development of the pandemic which focused on reducing stress based on mindfulness and a cognitive reframing of teacher well-being. The intervention improved resilience, mindfulness, and both subjective and psychological well-being of teachers during the first confinement, while in the control group, higher levels of burnout and decreased well-being were found [31]. In China, a multicomponent intervention based on positive psychology and the Positivity, Relationship, Outcomes, Strength, Purpose, Engagement, and Resilience (PROSPER) framework of well-being outcomes had significant effects on the domains of teacher well-being; positivity, resilience, strengths, and purpose, as tools for improving teacher outcomes, thrived in the development of the COVID-19 pandemic [32]. Similarly, in another intervention with Uruguayan teachers based on positive psychology and character education, the results showed improvements in psychological well-being and self-efficacy.

Psychological distress decreased in the participants alongside the strengthening of didactic pedagogical competencies for the application of positive psychology applied in educational centers [33]. Finally, Adelaide's character, resilience, and well-being online masterclass program is an initial teacher training program for Bachelor of Education students, consisting of a series of masterclasses with topics on positive psychology, teacher well-being, character strengths, professional teaching experience, and other relevant topics [34].

Research during and post COVID-19 has shown that we cannot sacrifice teachers; the education system must also serve teachers in their career development, their retention in the system, the prevention of psychosocial risks, and the promotion of teachers' well-being and health. In addition, it is essential to consider the objective rewards of salary and work benefits and psychosocial rewards in the design of teacher–public policies that ensure teacher job retention in a context of personal and professional growth [35]. The

educational crisis brought about by COVID-19 has generated both psychosocial risks and growth opportunities for teachers. These are challenges to address from the perspective of sustainable development in its SDG 3 Health and Well-being and SDG 4 Quality Education. Changes in education systems should address teacher well-being, socioemotional competencies, relationships in the educational center, and resilience as essential elements in the construction of sustainability through the improvement of teachers' working conditions, teaching practices, and educational quality [36]. Educational policies should emphasize teacher professional development, understood as a complex, continuous, evolutionary, and multidimensional phenomenon. Teacher training should be at the center of the actions undertaken to achieve changes and sustainability in education; these programs should encompass the educational center as a learning context, the contents should be meaningful and contextualized within the needs of the education center, accounting for collaborative problem-solving among professional peers, the impact on student learning, the consolidation of networks among teachers, and its integration with didactics, content programs, and educational management. In this sense, several ways to carry out these instances of teacher professional development have been implemented in the world: training programs, learning circles, learning communities, and pedagogical expeditions [37].

During the pandemic, a training experience in character education was carried out for principals and teacher coordinators with the dual purpose of developing professional competencies for the application of positive psychology in educational centers, as well as strengthening teacher well-being—specifically, their dedication and enthusiasm, together with personal resources such as self-efficacy and resilience.

The research questions that guided this process were: (a) What effects can the multicomponent intervention program have on teacher well-being, measured by dedication and enthusiasm, on the personal resources of self-efficacy and resilience, as well as on teacher training? (b) What learning generated in this training program strengthened professional teaching competencies with respect to the implementation of positive psychology applied to education in their schools?

Thus, the research objectives were: (1) to determine changes in teacher well-being (enthusiasm and dedication), personal resources (self-efficacy and resilience), and teacher education of the participants after the application of the program; (2) To determine if there are differences, after the intervention, in the scores the teacher well-being (enthusiasm and dedication), personal resources (self-efficacy and resilience), and in teacher education according to moment, degree, and teaching experience. In order to respond to these objectives, the professional development of teachers is promoted through the strengthening of specific competencies related to the management of positive psychology as applied to education in their schools.

## 2. Materials and Methods

### 2.1. Type of Research and Design

The present research was developed following a quasi-experimental design with measurements before and after the intervention (pre-test–post-test) in a single group [38].

### 2.2. Participants

Thirty-two people participated in the sample; they were teaching coordinators and school principals, 97.50% ($n = 30$) of whom identified themselves as women, and 6.25% ($n = 2$) as men. The age ranged from 29 to 58 years, with a mean of 45.97 and a standard deviation of 7.39 (CV = 16.08%). Regarding the degree (university degree), 59.38% ($n = 19$) had a primary school teacher's degree and 37.50% ($n = 12$) had a secondary or high school teacher's degree. Only 3.13% ($n = 1$) reported having a technical degree in teaching. The experience was distributed as follows: 28.12% ($n = 9$) of the persons had been working as teachers for periods of 5 to 16 years, and 71.88% ($n = 23$) for a time span ranging from 17 to 21 years.

### 2.3. Instruments

2.3.1. General Self-Efficacy Scale (GSES)

The Spanish version of the general self-efficacy scale was administered [39]. This measure is composed of 10 items that refer to a single dimension: a construct representing the person's perception of his or her capabilities and resources. Each item is configured in a Likert style with five options: 1 (never), 2 (sometimes), 3 (frequently), 4 (almost always), and 5 (always). Psychometric aspects were evaluated during the adaptation of the scale, finding evidence of validity and reliability [39]. Finally, Cronbach's alpha and McDonald's omega coefficients were calculated to measure the reliability of the questionnaire, detecting values of $\alpha = 0.75$ and $\omega = 0.70$, respectively.

2.3.2. Resilience, Dedication, and Enthusiasm Scales

These scales were obtained from the questionnaire used in the Study of Teachers' Occupational Health in Uruguay (ESODU), research undertaken by the National Institute for Educational Evaluation to study the mental health and well-being of teachers at the national level [29]. For its creation, a rigorous process of selection and adaptation of several pre-existing scales was carried out through the implementation of quantitative and qualitative techniques. The validation of the final instrument was undertaken after choosing a pilot sample and obtaining data with which aspects such as factorial, convergent, discriminant, and reliability validity were determined. For the research developed here, four items from the resilience section were used based on the scale originally proposed by Connor and Davidson [40]. Likewise, the three items of the work dedication and enthusiasm section were taken into account, which was constructed based on the HERO questionnaire [41]. The scale items are configured in a Likert format with four response options: 1 (strongly disagree), 2 (disagree), 3 (agree), and (4) strongly agree. The resilience items reached Cronbach's alpha $\alpha = 0.70$ and McDonald's omega $\omega = 0.73$. The dedication and enthusiasm items reached Cronbach's alpha $\alpha = 0.81$ and McDonald's omega $\omega = 0.82$.

2.3.3. Teacher Training Scale

Teacher training was evaluated with a scale designed especially for the project (ad hoc). This instrument is made up of eight items configured on a Likert scale with four response options: 1 (strongly disagree), 2 (disagree), 3 (agree), and 4 (strongly agree). In turn, these items are grouped into two dimensions: (a) teaching attitude towards promoting mental health and well-being; (b) teaching coordination and educational management of resources in positive psychology applied to education. Expert judgment evaluated content validity, showing clarity, relevance, systematization, and theoretical adequacy. Likewise, a confirmatory factor analysis was performed, which reported adequate figures and allowed confirming the proposed structure. The convergent and discriminant validity was also examined, a procedure that reported satisfactory results. Finally, Cronbach's alpha and McDonald's omega coefficients were calculated to measure the reliability of the questionnaire, detecting values ranging from $\alpha = 0.69$ to $\omega = 0.88$.

### 2.4. Program

The training program in character education and positive psychology was aimed at teachers with educational management functions: either directors or teaching coordinators of educational institutions in the Eastern Republic of Uruguay. The program was a theoretical–practical training program in the field of character education and the development of socioemotional competencies based on the contributions of positive psychology, with a duration of about 50 direct hours of work that was carried out in the context of the COVID-19 crisis to incorporate innovative practices into their educational management and promote their well-being. The online work sessions were synchronous and practical activities were carried out asynchronously, the latter oriented to the development of practices with implications for personal growth and teacher professional development.

The sustainable development of education and the promotion of mental health and well-being require teachers who are trained, qualified, and committed to educational change. They consisted of synchronous work sessions to develop the training modules of the program, which were theoretical and practical: (a) introduction to positive psychology applied to character education: areas of application and the development of well-being in the classroom; (b) PERMA model of well-being and importance of positive emotions; this model understands well-being as a multidimensional construct, composed of Positive Emotion, Engagement, Relationships, Meaning, and Accomplishment (hence PERMA); (c) interventions in applied positive psychology: personal growth, character strengths, and resilience; (d) mindfulness: implications, exercises, and applications at personal and educational levels; (e) humanity strengths and interpersonal relationships; (f) positive psychology and constructs of self-esteem and self-efficacy: personal and educational implications; (g) leadership as a character strength and types of leadership in educational management; (h) fixed and growth mindset; (i) emotional strengths: courage, perseverance, frustration tolerance, and others; (j) transcendence strengths: emphasis on gratitude; (k) cognitive or wisdom strengths. There was a great variety of strategies in the development of the program. However, we would like to mention the following: design and execution of strategies, mindfulness dynamics, character strengths, management of emotions, communication, and others, in addition to projecting the projects of the educational center for the following year.

The cross-cutting theme of the program was educational management as the context of application at the work level and personal growth focused on strengthening personal resources at the individual level. The participants referred in their reasons for entering the program: (a) teacher training in character education and well-being; (b) updating and professional development to improve the quality of education and management. Thus, personal, professional, and school management applications were discussed throughout the training modules. A teacher coordinator participated in all the meetings as a way of creating a common thread to generate links and detect the needs of the group, accompanying the expert teachers of the aforementioned modules. All the meetings provided a space for the exchange of experiences and the generation of ideas, which was highly valued by all the teachers as a way of learning among colleagues, and a communication group and online platform were created to host the recorded classes, resources, and materials.

The research was conducted following the ethical guidelines of the APA, Helsinki declaration, approval of the ethics committee of Jóvenes Fuertes Uruguay (190422/April 2019), and informed consent that contained the objective of the research, purpose, and usefulness of the research. Pre-application and post-application measurements of the training program were performed as an approximation to measure changes in the variables.

*2.5. Procedure*

The program was in charge of Jóvenes Fuertes Uruguay and was conducted online as a response to accompany teachers in the development of the COVID-19 crisis. The sample selection was made by one of the program's funding entities (Reaching U), which aims to generate educational leaders capable of facing the challenges of the educational system to improve the quality of education. Therefore, the intervention was carried out in a natural group of previously constituted teachers, without the option of having a control group; the purpose of the foundations involved was to accompany the teachers in this educational transition as part of a call to action; the search for control groups would have delayed the process of starting the program. The research was conducted following the ethical guidelines of the APA, Helsinki declaration, approval of the ethics committee of Jóvenes Fuertes Uruguay (190422/April 2019), and informed consent that contained the objective of the research, purpose, and usefulness of the research. Pre-application and post-application measurements of the training program were performed as an approximation to measure changes in the variables. Data collection was performed employing the psychometric scales described above and was carried out by the program's teaching coordinator after

explaining the informed consent and ethical aspects of participation. An online form with sociodemographic data and the corresponding scales was completed and applied before and after the intervention.

### 2.6. Data Analysis

The statistical analysis was developed using the most recommended approaches for data from one-group pre-test–post-test designs: analysis of covariance (ANCOVA) and repeated measures analysis of variance (MR ANOVA) [42–45]. As such, the dependent variables used in the ANCOVA were the scores of all psychological constructs after the intervention, while the intersubject factors were teaching qualification and years of experience. Gender was not used due to the imbalance observed in the sample on this characteristic, as only 6.25% (*n* = 2) of the participants identified themselves as male. Regarding the MR ANOVA, the dependent variables were the pre-test and post-test scores, while the covariates were age, teaching degree, and years of experience. The last two aspects were not used as intersubject factors because the research was not focused on analyzing interaction terms but on measuring longitudinal changes in the psychological variables analyzed.

Consequently, the assumptions necessary to perform the analysis of covariance were checked. For this purpose, the homogeneity of regression slopes was inspected through the significance of the interaction between intersubject factors and covariates, while homoscedasticity was validated by Levene's test [46,47]. Furthermore, linearity between covariates and dependent variables according to intersubject factor levels was tested by constructing scatter plots. Moreover, the residual normality assumption was examined by implementing the Shap andest, but also by constructing quantile–quantile plots (Q–Q plots). Furthermore, the presence of outliers and influence points was inspected using robust Mahalanobis distances, standardized residuals, and Cook's distances [42,46]. Considering that no significant violations were detected during this phase, the decision was made to proceed with the analysis as initially planned.

The parametric assumptions of the repeated measures ANOVA were also tested. Therefore, it was verified by Box's M test that the variance–covariance matrices did not differ significantly according to the groups defined by the intersubject factors [42,44,45]. The sphericity assumption was not tested because only two temporal measurements were performed, while the remaining hypotheses were tested employing the same tools used to validate the ANCOVA conditions [42–47]. Ultimately, psychological variables were presented as means and standard errors, while age was described through mean and standard deviation. Sociodemographic characteristics were presented tabularly using frequencies and percentages. The effect size was established with the partial eta coefficient squared, classifying it as low ($\eta^2 > 0.01$), medium ($\eta^2 > 0.06$), or high ($\eta^2 > 0.14$), while significance was set for figures less than 0.05. Finally, the IBM SPSS 27 program performed data processing and analysis.

## 3. Results

### 3.1. Variables of Interest in the Baseline

The variables of interest at baseline are shown in Table 1. As can be seen, participants exhibited an average level in each of the constructs analyzed, and no significant differences were found when compared according to gender, degree, or years of experience.

### 3.2. Effect of the Intervention on the Variables of Interest

The results of the intervention effect are presented in Table 2. The RM ANOVA suggests that the program improved self-efficacy, resilience, dedication, enthusiasm, and teacher training. Concerning self-efficacy, significant differences were found between the pre-test and post-test scores, changing from 32.85 (SE = 0.98) to 37.57 (SE = 1.18), indicating a large effect (F = 18.17, *p* < 0.001, $\eta^2 = 0.40$). Resilience increased from 11.85 (SE = 0.56) to 14.44 (SE = 0.64), also revealing a large effect (F = 13.41, *p* < 0.01, $\eta^2 = 0.33$). The dedication and enthusiasm showed significant growth. In this case, the scores improved from 9.58

(SE = 0.59) to 11.63 (SE = 0.41), showing a large effect size (F = 8.09, $p < 0.01$, $\eta^2 = 0.23$). For the teacher training, the pre-test and post-test scores were 28.43 (SE = 0.77) and 30.91 (SE = 0.54), revealing a large effect (F = 8.36, $p < 0.001$, $\eta^2 = 0.24$). On the other hand, the ANCOVA revealed that there were no differences in the post-test scores according to degree and teaching experience.

**Table 1.** Description of the variables of interest before the intervention (pre-test).

| Variables | According to Qualification | | | According to Years of Experience | | | Total |
|---|---|---|---|---|---|---|---|
| | PST [a] | HST [b] | Level | 5–16 | 17–21 | Level | |
| Self-efficacy | 34.21 (2.51) | 33.31 (2.95) | Medium | 32.89 (2.57) | 34.22 (2.70) | Medium | 33.84 (2.69) |
| Resilience | 12.26 (1.48) | 12.15 (1.41) | Medium | 11.89 (1.90) | 12.35 (1.23) | Medium | 12.22 (1.43) |
| Dedication and Enthusiasm | 10.21 (1.44) | 10.31 (1.70) | Medium | 10.11 (1.17) | 10.30 (1.66) | Medium | 10.25 (1.52) |
| Teacher training | 34.21 (2.51) | 33.31 (2.95) | Medium | 32.89 (2.57) | 34.22 (2.70) | Medium | 33.84 (2.69) |

The mean (unadjusted) is shown and in parentheses the standard deviation. The level corresponds to the qualitative interpretation obtained through the scales of each scale. Means with the same subscript do not differ statistically at the 0.05 level. ([a]) The "Primary school teacher", ([b]) The category high school teacher's degree.

**Table 2.** Evolution of the variables of interest before and after the program and comparison of the average of the variables of interest according to degree and experience.

| Variables | According to Moment (RM ANOVA) | | | According to Qualification (ANCOVA) | | | According to Years of Experience (ANCOVA) | | |
|---|---|---|---|---|---|---|---|---|---|
| | Pre | Post | F ($\eta^2$) | PSTa | HSTb | F ($\eta^2$) | 5–16 | 17–21 | F ($\eta^2$) |
| Self-efficacy | 32.85 (0.98) | 37.57 (1.18) | 18.17 *** (0.40) | 37.74 (1.25) | 38.60 (1.12) | 0.65 [†] (0.02) | 38.86 (1.27) | 37.47 (1.13) | 1.48 [†] (0.05) |
| Resilience | 11.85 (0.56) | 14.44 (0.64) | 13.41 ** (0.33) | 14.47 (0.73) | 14.67 (0.64) | 0.10 [†] (0.01) | 14.17 (0.73) | 14.98 (0.66) | 1.54 [†] (0.05) |
| Dedication and Enthusiasm | 9.58 (0.59) | 11.63 (0.41) | 8.09 ** (0.23) | 11.92 (0.51) | 11.43 (0.44) | 1.31 [†] (0.05) | 11.71 (0.50) | 11.64 (0.46) | 0.03 [†] (0.01) |
| Teacher training | 28.43 (0.77) | 30.91 (0.54) | 8.36 *** (0.24) | 31.31 (0.64) | 30.66 (0.55) | 1.43 [†] (0.05) | 30.81 (0.63) | 31.16 (0.58) | 0.40 [†] (0.01) |

Adjusted marginal means and standard errors are presented (in parentheses). *F*-statistics and effect sizes are also presented using the partial eta-squared coefficient (in parentheses). [†] $p > 0.05$. ** $p < 0.01$. *** $p < 0.001$.

### 3.3. Outcomes at the Level of Teacher Professional Development, Character Education, and Well-Being

The participants' intentions for enrollment in the training program were classified into two dimensions: (a) teacher training in character education and for well-being; and (b) updating, teacher training, educational quality, and leadership management. The participants designed techniques, activities, and strategies based on the learning generated in the modules; for example, character strengths, positive emotions, communication, assertiveness, and well-being; they also carried out or at least planned for the following academic year's educational projects and center campaigns focused on health promotion, including improvement of relations among teaching peers, gratitude, emotional regulation, optimism, and others. Likewise, participants report having expanded their resources at the level of communication, mindfulness, leadership, emotional intelligence, and self-efficacy for the management of teaching work; it was observed that participants had the didactic capacity to begin to integrate learning according to the contents and programs taught in schools. The participants report that one of the valued strengths of this group training had been sharing practical experiences and considering the course as a space for professional learning and caring for the well-being and health of teachers. A few months after the end of the training, a follow-up meeting was held in the middle of the academic year, led by the coordinator who participated in all the modules to explore the applications that were being carried out in the educational center, and to have organized spaces to share with other teacher colleagues what they learned in training in their teaching or management coordinations. They also reported that the topics of the modules studied that had more significant applicability in their educational centers were: (a) assertive communication and

relationships; (b) emotional strengths; (c) resilience and gratitude. In closing, the group of teachers expressed a willingness to maintain a learning network among colleagues.

## 4. Discussion

This work had two objectives that guided the process: (1) to determine changes in the teacher well-being (enthusiasm and dedication), personal resources (self-efficacy and resilience), and teacher education of the participants after the application of the program; (2) to determine if there are differences, after the intervention, in the scores in the constructs teacher well-being (enthusiasm and dedication), personal resources (self-efficacy and resilience) and teacher education according to moment, degree and teaching experience. The results suggest that there were statistically significant increases in the personal resources of the participating teachers, namely, self-efficacy, resilience, dedication, and enthusiasm (teacher well-being), and in teacher training. In addition, it is interpreted that they increased their beliefs in their abilities in planning, organizing, and directing, as required to meet specific goals or objectives, i.e., they perceive themselves as more effective for their professional tasks [39].

Similarly, there were increases in resilience, expressed as the ability to overcome traumatic events or maintain adequate performance despite the adverse context. Certainly, the COVID-19 pandemic was considered a psychosocial trauma [40]. Likewise, teacher well-being consisted in the dedication and enthusiasm indicators of engagement which had increased. Participants reported increases in their positive work-related mental state expressed in energy, dedication, and enthusiasm in their professional activities [29,41]. Concerning the promotion of teachers' professional development, there are specific improvements in teachers' attitudes toward the promotion of mental health and well-being, as well as in teaching coordination and the management of educational resources in positive psychology applied to education [48].

The improvement of personal resources and professional teacher training can account for the effectiveness of multicomponent intervention based on positive psychology applied to education. It is congruent with other experiences of intervention with teachers reported before the pandemic [49–55], as well as with interventions conducted during the COVID-19 pandemic with teachers [31,32,34,48], and even with other intervention studies conducted with the general population during COVID-19 [56–58]. The results also correlate with the effectiveness of multicomponent positive psychology-based interventions conducted in general or nonclinical populations [59–62], which report small to moderate magnitude effect sizes. The results of the present study are congruent with precedents that have control groups or placebos in their research design; however, this was an intervention carried out in the midst of the COVID-19 pandemic, in which the intention was to accompany teachers during this educational emergency [49]. The results are quite optimistic for the development of future interventions with teachers; however, it is important to take into account the limitations of the research design executed.

The results also point to the importance of personal resources on teacher well-being and health that have been highlighted by studies conducted in the educational emergency generated by the pandemic, emphasizing self-efficacy, resilience, and dimensions of engagement [17–19,21–24,28,35,63–66]. With respect to professional development and teacher training, the results are consistent with the findings related to the need perceived by teachers for training instances aimed at their well-being and health, especially emphasizing collaborative work, opportunities to share with other teaching peers, and the exchange of good practices among peers from the same school or others close to the school community [20,25–27]. In this sense, in Uruguay, the National Institute for Educational Evaluation has found that among the success factors linked to schools with better results than expected is the presence of a strong professional teaching community that offers support, guidance, and collaborative leadership [27]. These are important factors related to social and instrumental support in the design of training experiences for teachers' professional development.

Among the practical–theoretical implications arising from this research are the following: contributions to psychological intervention based on positive psychology for the improvement of psychosocial resources, well-being, and health in teachers, these findings are of relevance to various theories, for example: the theory social cognitive [67], Job Demands-Resources Model [68], the classification of character strengths and virtues [69], resilience [40], multidimensional well-being [70,71] and engagement as an approach to teacher well-being in the dimensions of dedication and enthusiasm [41]. All of these contribute to explaining psychosocial functioning in work contexts and can provide theoretical foundations for the design of psychosocial interventions based on specific or multi-component concepts of mental health such as cognitive behavioral models, health psychology, or positive psychology to prevent teacher burnout and even attrition in aid of the promotion of teacher retention to the educational system [72]. The findings make us reflect on the relevance of integrated theoretical approaches to explain teacher well-being and health from the perspective of personal, occupational, and contextual resources oriented towards the development of professional teaching competencies [73–76].

Similarly, our results are congruent with the experience of training based on positive education with teachers in Finland, who were trained in a program with an emphasis on character strengths, PERMA well-being, positive emotions, enhanced learning through positive emotions, hope, resilience, and a positive mindset. In addition to increases in participants' well-being, teachers reported the applicability of what they had learned to different situations in the school, as well as contributing to personal growth and teacher professional development, peer-to-peer teacher exchanges linked to social support and knowledge sharing. They also noted the applicability of their training to challenges related to teachers' attitudes towards this type of training, work overload, and improvement of relational skills in the school [77].

On the other hand, the ANCOVA revealed that there were no differences in the posttest scores according to degree and teaching experience. These results differ from the findings found by Muñoz and Correa [78], who report that personal variables of teachers, such as years of experience, could be an explanatory factor of resilience or how they adapt and cope with the challenges of their environment. Thus, we believed that years of experience could act as a positive aspect given the security provided by work experience in developing personal traits such as resilience, well-being, and self-efficacy in teachers [79,80]. However, in this case, confirming these aspects was impossible.

In addition to the direct implications for the personal growth and professional development of teachers, training instances in social–emotional competencies are not only effective in promoting the well-being and health of teachers; they also have an impact on teaching practices, seeking to improve the teaching and learning experience in conjunction with the academic performance of their students through the application of strategies and techniques learned in social–emotional competencies training [81].

One of the study's main limitations is the reported research design that lacks a control group and is carried out with a small sample, which raises issues concerning the generalization of the conclusions, even cause–effect associations, and other issues related to the internal validity typical of quasi-experimental studies. However, the results are encouraging because they were conducted in a social, historical moment marked by COVID-19 adversity. They also demonstrate a theoretical consistency and congruence with other studies with similar effect sizes, control groups, and other advanced criteria of experimental research designs [33,59–62]. The study is also congruent with other interventions that did not have control groups or placebos in their research designs [49].

Despite the limitations of the research design and the small sample of participants, this study has some strengths. Among them, we demonstrate that multicomponent interventions in positive psychology could be devices for the promotion of personal and work resources that function as protective factors for the well-being and health of teachers, given the characteristics of the environments (VUCA: volatility, uncertainty, complexity, ambiguity) [63]. Similarly, this research represents an advance in the line of studies of

positive psychology interventions in countries for non-Western, Educated, Industrialized, Rich and Democratic (WEIRD) contexts. It is an effort to apply positive psychology and education for the transformation of our societies in the midst of uncertainty, abrupt changes, crisis, and the expectations of educational recovery generated post COVID-19 [32,48,59].

It is highly recommended to design, implement, and evaluate training instances aimed at updating teaching competencies, health promotion, and prevention of the main psychological labor risks associated with teaching; it is important to retain good teachers in the educational system who can respond to the needs of students and train them for the challenges of the present and the future. Future studies may include methodological strengths in the research design: control groups, follow-up measures after the end of the program to analyze the behavior of variables sensitive to the intervention, and a more homogeneous sample at the gender level (although teaching has a higher profile of women than men) [29]. On the other hand, technologies can be allies for the development of mass interventions for the promotion of teachers' well-being and health; for examples, massive open online courses (MOOC) for the design of learning experiences that are means for teachers' professional development. It is vital that these instances, whether online, semi face-to-face, or entirely face-to-face, include in the instructional design group activities that permit participants to create teachers' work networks, learning circles, and even teachers' professional learning communities [6]. In future research, variables related to the style of educational management carried out by principals and teaching coordinators, such as leadership style, compensation, the recognition system of each school, means conflict resolution, and the diversity management in the school, can be considered in the design of the research. We may also explore which personal, professional, or contextual characteristics of the school promote (or do not promote) the incorporation of positive psychology as it pertains to educational or teaching management.

## 5. Conclusions

The training program based on a multicomponent intervention in positive psychology applied to education generated essential changes in teacher well-being and health during the COVID-19 pandemic: specifically, statistically significant increases in enthusiasm, dedication, self-efficacy, resilience, and teacher training. There were improvements in teacher well-being, personal resources, and teacher training after the application of the training program. It can also be concluded that this program is a professional collaborative opportunity among teaching peers which enabled teachers' professional development by strengthening specific competencies related to the management of positive psychology applied to education in their schools. However, these conclusions should not be generalized due to limitations in the pre- and post-research design, with only one group.

**Author Contributions:** Conceptualization, D.G.-Á. and M.J.S.; data curation, D.G.-Á. and J.H.-L.; formal analysis, D.G.-Á., R.C.-R. and J.H.-L.; funding acquisition, M.J.S. and D.G.-Á.; investigation, D.G.-Á. and R.C.-R.; methodology, D.G.-Á. and M.J.S.; project administration, M.J.S. and J.H.-L.; resources, M.J.S. and J.H.-L.; software, D.G.-Á., M.J.S. and R.C.-R.; supervision, D.G.-Á. and R.C.-R.; validation, D.G.-Á., M.J.S. and R.C.-R.; visualization, R.C.-R.; writing—original draft, D.G.-Á., M.J.S. and R.C.-R.; writing—review and editing, D.G.-Á., M.J.S., R.C.-R. and J.H.-L. All authors have read and agreed to the published version of the manuscript.

**Funding:** This research was funded by Jóvenes Fuertes Uruguay and Fundación ReachingU. The APC was funded by Universidad Metropolitana, Venezuela, Universidad de Montevideo, Uruguay and Asociación Jóvenes Fuertes Uruguay.

**Informed Consent Statement:** Informed consent was obtained from all subjects involved in the study.

**Data Availability Statement:** Further inquiries can be directed to the corresponding author.

**Acknowledgments:** To Jóvenes Fuertes Uruguay, the specialist teachers who carry out the intervention, Carina Zerbino, and Inés Ruiz for their support in the project.

**Conflicts of Interest:** The authors declare no conflict of interest.

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
