# Peer review of "Teacher Professional Development, Character Education, and Well-Being: Multicomponent Intervention Based on Positive Psychology"

_sustainability, doi:10.3390/su15139852_

Round 1

Reviewer 1 Report

The manuscript is interesting and deals with a current and important issue - teacher professional development, character education and well-being enhanced by appropriate training to prepare teachers for a possible future crisis, or other types of educational challenges.

lines 15 and 142-143

It would be nice (not necessarily in the summary), even if it is considered well known, to explain the meaning of the abbreviations of the UN Sustainable Development Goals SD3 and SD4.

lines 187-206

In paragraph 2.3, the order of descriptions of scales/instruments could be aligned with the order in Tables 1 and 2, or vice versa.

lines 215-219

In the resilience section it should be explained that in Connor and Davidson scale each item is configured in a Likert style with five options, like it is explained in other sections. Also, items configuration from HERO questionnaire used in dedication and enthusiasm section should be explained to.

line 237

It is not quite clear is it “synchronous work sessions of practical theoretical modules” or maybe “synchronous work sessions of practical and theoretical modules”?

line 239

It would be useful to list (in parentheses) words that explain the abbreviation PERMA, and thus to some extent the model itself.

line 271

The order of the letters RM (Repeated measures in RM ANOVA) was probably accidentally permuted into MR.

line 315-318

This seems to be a redundant repetition of data, because it is once again listed in the text that follows, where it is purposeful.

line 323

Even if the words dedication and commitment are synonyms, it would be appropriate to use the same word  as in the Table.

I believe that the manuscript can be slightly improved and then accepted for publication in Sustainability.

Author Response

Dear reviewer

Best regards!

Thank you very much for your efforts and for all the suggestions and indications made to improve our work.

The following are the responses to this latest review in the hope that we will be able to meet your expectations to your satisfaction,

The authors

Revierwer Report

lines 15 and 142-143: It would be nice (not necessarily in the summary), even if it is considered well known, to explain the meaning of the abbreviations of the UN Sustainable Development Goals SD3 and SD4.

Answer: We have described the meaning of the abbreviations of the Sustainable Development Goals. 

lines 187-206: In paragraph 2.3, the order of descriptions of scales/instruments could be aligned with the order in Tables 1 and 2, or vice versa.

Answer: Thank you very much for your suggestion, we have modified the order of presentation of the questionnaires in the method section to match the order of presentation of the tables.

lines 215-219: In the resilience section it should be explained that in Connor and Davidson scale each item is configured in a Likert style with five options, like it is explained in other sections. Also, items configuration from HERO questionnaire used in dedication and enthusiasm section should be explained to.

Answer: Thank you very much for the suggestion, we have modified the description of the questionnaires used in the research, including the requested information.

line 237: It is not quite clear is it “synchronous work sessions of practical theoretical modules” or maybe “synchronous work sessions of practical and theoretical modules”?

Answer:  Thank you very much for your suggestion, we have expanded the description of the development and implementation of the intervention program to respond to this request.

line 239: It would be useful to list (in parentheses) words that explain the abbreviation PERMA, and thus to some extent the model itself.

Answer: we include the description of the acronym PERMA

line 271: The order of the letters RM (Repeated measures in RM ANOVA) was probably accidentally permuted into MR.

Answer: Thank you very much. We have made the corresponding modification.

line 315-318: This seems to be a redundant repetition of data, because it is once again listed in the text that follows, where it is purposeful.

Answer: Thank you very much for your comment. We have removed the redundant information.

line 323: Even if the words dedication and commitment are synonyms, it would be appropriate to use the same word  as in the Table.

Answer: we have included the same terms from the tables in the manuscript. 

Reviewer 2 Report

Overall Comments:

The reviewer applauds the researchers on this evaluation study. The introduction is well-sourced and appropriately leads to the need for teacher/leader professional development on this topic, as well as the need to evaluate the impacts of professional development. The design of the study is sound, given the limitations described. The reviewer appreciates the use of scales to measure constructs, with attention given to reporting of reliability and validity (please double-check that the specific reliability is reported for all scales, e.g., 2.3.3). The data is well presented in table format. Discussion, conclusions, and recommendations align with the study but have opportunities for improvement via some revision.

Opportunities to Improve:

There are many areas where grammatical improvements should be made. Run-on sentences are a concern, as well as some areas where commas should be replaced by periods. The full manuscript needs an in-depth review prior to publication.  

For line 26, remove “can be an instance” and replace it with “training programs can improve teacher health and well-being…”

Please consider revising objective 2. Promoting should not be a research objective. The results of the study could be used to promote, etc. The central hypothesis is also unclear and it is recommended to remove it. In my opinion, the hypothesis statement is really more about the purpose of the program/study. The objectives of this research should really reflect the evaluative nature of this study. For example, “Determine participant changes to x, x, x, (constructs) based upon the professional development intervention..” Also, it appears that a major portion of your results compares teacher characteristics (teaching level, years of experience) on intervention outcomes. This could also be a research objective (e.g., Determine if participants’ qualifications and prior years of experience impacted intervention outcomes). These examples would need to be revised, but illustrate my point.

Line 178 should be 93.7%, not 97.5%

Lines 181-182: Is the degree their teaching level or their educational degree? This may be unclear to global readers.

Line 182. Change to “only one participant reported having a technical…”

Line 186: Need some discussion on the survey itself prior to the discussion of the instruments. When and how was it distributed?

Could be beneficial to move 2.4 Program and procedure prior to materials and methods. The materials and methods section should really describe how the evaluation of the program took place. Was the program synchronous or asynchronous? I would recommend using subheadings in this section. In my opinion, this is all very important information. As currently written, sections seem duplicative and unclear. Please reorganize this to make it succinct and to improve clarity.  

Revise 4.0 Discussion objectives. Again, I would imagine the research objectives of your study were based on the evaluation of the intervention. To determine or to assess the impacts of the program.. the objectives/goals of the “program” were to improve or to promote, but not the objectives of the research.

A good portion of the results describes the findings between qualification and years of experience on intervention impact. However, the non-findings between the groups were largely glossed over in the discussion, conclusions, and recommendations sections. I would add more on these areas within these sections to align with your investigation and findings.

Thank you. This paper and interesting and informative.

Author Response

Dear reviewer

Best regards!

Thank you very much for your efforts and for all the suggestions and indications made to improve our work.

The following are the responses to this latest review in the hope that we will be able to meet your expectations to your satisfaction,

The authors

Review Report

The reviewer appreciates the use of scales to measure constructs, with attention given to reporting of reliability and validity (please double-check that the specific reliability is reported for all scales, e.g., 2.3.3).

Answer: Thank you very much for your input, we performed the corresponding calculations for the reliability analysis, the results have been included in the description of each instrument.

Answer: Thank you very much for the suggestion, we include the calculations of the feasibility of the identified scales under study. These were reported in the description of each questionnaire in the method.

For line 26, remove “can be an instance” and replace it with “training programs can improve teacher health and well-being…”

Answer: We have modified the wording of this information.

Please consider revising objective 2. Promoting should not be a research objective. The results of the study could be used to promote, etc. The central hypothesis is also unclear and it is recommended to remove it. In my opinion, the hypothesis statement is really more about the purpose of the program/study. The objectives of this research should really reflect the evaluative nature of this study. For example, “Determine participant changes to x, x, x, (constructs) based upon the professional development intervention..” Also, it appears that a major portion of your results compares teacher characteristics (teaching level, years of experience) on intervention outcomes. This could also be a research objective (e.g., Determine if participants’ qualifications and prior years of experience impacted intervention outcomes). These examples would need to be revised, but illustrate my point. 

Answer: Thank you very much for your comment. We have modified the research questions and objectives presented to better adjust them to the purposes and analysis offered in the manuscript. 

Line 178 should be 93.7%, not 97.5%

Answer: Thank you very much we have made the modification.

Lines 181-182: Is the degree their teaching level or their educational degree? This may be unclear to global readers.

Answer: Thank you very much, the term we kept in the document was "educational degree".

Line 186: Need some discussion on the survey itself prior to the discussion of the instruments. When and how was it distributed?

 Answer: Thank you very much we have made the modification. This aspect was clarified in the procedure.  

Could be beneficial to move 2.4 Program and procedure prior to materials and methods. The materials and methods section should really describe how the evaluation of the program took place. Was the program synchronous or asynchronous? I would recommend using subheadings in this section. In my opinion, this is all very important information. As currently written, sections seem duplicative and unclear. Please reorganize this to make it succinct and to improve clarity.  

Answer: Thank you very much for the suggestion. We rewrote the procedure paragraph to make this and other aspects of the research clearer. 

Revise 4.0 Discussion objectives. Again, I would imagine the research objectives of your study were based on the evaluation of the intervention. To determine or to assess the impacts of the program.. the objectives/goals of the “program” were to improve or to promote, but not the purposes of the research.

  Answer: Answer: Thank you very much for the suggestion. We have modified the wording of the objectives of the research.

A good portion of the results describes the findings between qualification and years of experience on intervention impact. However, the non-findings between the groups were largely glossed over in the discussion, conclusions, and recommendations sections. I would add more on these areas within these sections to align with your investigation and findings.

Answer: Thank you very much for the suggestion, we have included an objective consistent with this result and include information about the finding in the discussion of the manuscript.

Reviewer 3 Report

Thank you for giving me an opportunity to review this interesting paper regarding the intervention effects of a character education training program in the Eastern Republic of Uruguay. By comparing the changes of teacher well-being during the online intervention program, the authors concluded that the training program was effective for improving teachers’ health, well-being, and professional development. The article lays out its argument clearly. Nevertheless, the marginal contributions might be mainly on the descriptive level. There are some problems that the authors should carefully address:

1. The study design was not suitable for evaluating the effectiveness of an intervention program. In fact, it is possible that the teacher well-being indicators changed naturally over time before and after the intervention. Although the authors were aware of the limitations of single group design, the possible consequences of lacking control group have not been discussed deeply enough in the current paper.

2. Teacher well-being is the main dependent variable in empirical analysis. However, the authors did not clearly define this important concept. If “self-efficacy, resilience, dedication and enthusiasm, and teacher training” are the operationalization of the dependent variable, relevant literature review should be provided with detailed explanations. In addition, the above variables did not directly involve health indicators, so the statement that the intervention has a promoting effect on teachers' health should be revised accordingly.

3. As shown in Table 2, there were no statistically significant differences between primary school teachers and high school teachers, or between 5-16 and 17-21 years of teaching experience groups (p>0.05). So, what are the implications of group comparison? The authors need to make further explanations in response to this question.

In summary, although the topic of this manuscript is attractive, the paper requires a Major Revision to beef up its methodological framing and explanation of results etc.

Author Response

Dear reviewer

Best regards!

Thank you very much for your efforts and for all the suggestions and indications made to improve our work.

The following are the responses to this latest review in the hope that we will be able to meet your expectations to your satisfaction,

The authors

Review Report

Thank you for giving me an opportunity to review this interesting paper regarding the intervention effects of a character education training program in the Eastern Republic of Uruguay. By comparing the changes of teacher well-being during the online intervention program, the authors concluded that the training program was effective for improving teachers’ health, well-being, and professional development. The article lays out its argument clearly. Nevertheless, the marginal contributions might be mainly on the descriptive level. There are some problems that the authors should carefully address:

  1. The study design was not suitable for evaluating the effectiveness of an intervention program. In fact, it is possible that the teacher well-being indicators changed naturally over time before and after the intervention. Although the authors were aware of the limitations of single group design, the possible consequences of lacking control group have not been discussed deeply enough in the current paper.

Answer: Thank you very much for your contributions, in the discussion section we will expand the analysis of these issues that are part of the limitations of the study. 

  1. Teacher well-being is the main dependent variable in empirical analysis. However, the authors did not clearly define this important concept. If “self-efficacy, resilience, dedication and enthusiasm, and teacher training” are the operationalization of the dependent variable, relevant literature review should be provided with detailed explanations. In addition, the above variables did not directly involve health indicators, so the statement that the intervention has a promoting effect on teachers' health should be revised accordingly.

Answer: Thank you very much for your contributions, we have expanded and rewritten on these aspects in the last part of the introduction. 

  1. As shown in Table 2, there were no statistically significant differences between primary school teachers and high school teachers, or between 5-16 and 17-21 years of teaching experience groups (p>0.05). So, what are the implications of group comparison? The authors need to make further explanations in response to this question.

Answer: Thank you very much for your contributions, we have expanded the analysis of this finding in the discussion of the manuscript. 

Reviewer 4 Report

- It's a very interesting topic with a strong design to consider for publication. However, I have a minor point before the process for the next step:

- The introduction needs to highlight the impact of covid-19 in general and especially on the educational field and try to highlight your research rationale with updated references. Here are some previous studies that have been conducted to evaluate the effect of covid-19 in different countries including education;

https://doi.org/10.3390/healthcare10101858

http://dx.doi.org/10.1136/bmjopen-2020-046006

https://doi.org/10.1016/j.sleepe.2022.100030

https://journals.sagepub.com/doi/full/10.1177/1179173X211053022

https://doi.org/10.3390/ejihpe12080079

https://doi.org/10.1371/journal.pone.0277368

https://doi.org/10.3389/feduc.2022.960660

Author Response

Dear reviewer

Best regards!

Thank you very much for your efforts and for all the suggestions and indications made to improve our work.

The following are the responses to this latest review in the hope that we will be able to meet your expectations to your satisfaction,

The authors

Review Report

- It's a very interesting topic with a strong design to consider for publication. However, I have a minor point before the process for the next step:

- The introduction needs to highlight the impact of covid-19 in general and especially on the educational field and try to highlight your research rationale with updated references. Here are some previous studies that have been conducted to evaluate the effect of covid-19 in different countries including education;

https://doi.org/10.3390/healthcare10101858

http://dx.doi.org/10.1136/bmjopen-2020-046006

https://doi.org/10.1016/j.sleepe.2022.100030

https://journals.sagepub.com/doi/full/10.1177/1179173X211053022

https://doi.org/10.3390/ejihpe12080079

https://doi.org/10.1371/journal.pone.0277368

https://doi.org/10.3389/feduc.2022.960660

Answer: Thank you very much for your contributions, which contribute to strengthen our work. We found the proposed research valuable and interesting and have included it in the introduction of our manuscript. 

Reviewer 5 Report

The conclusion are not clear.  A part of the conclusions chapter must be moved in disscusion section. In the conclusions should be remain only 2-3 clear ideeas.

Author Response

Dear reviewer

Best regards!

Thank you very much for your efforts and for all the suggestions and indications made to improve our work.

The following are the responses to this latest review in the hope that we will be able to meet your expectations to your satisfaction,

The authors

Review Report

The conclusión are not clear.  A part of the conclusions chapter must be moved in disscusion section. In the conclusions should be remain only 2-3 clear ideas.

Answer: Thank you very much for your input, we rewrote the conclusions, considering a more concise presentation, the rest of the relevant information has been included in the discusión section. 

Round 2

Reviewer 3 Report

I am satisfied with the work done by the authors, who addressed in a straightforward way all my comments. I think that, all things considered, this article now can be considered for publication.

Reviewer 5 Report

The article is well done